# On the relationship between Pathogenic Potential and Infective Inoculum

**Daniel F. Q. Smith**[ORCID]**, Arturo Casadevall**[ORCID]*

W. Harry Feinstone Department of Molecular Microbiology and Immunology, The Johns Hopkins Bloomberg School of Public Health, Baltimore, Maryland, United States of America

* acasade1@jh.edu

**Data Availability Statement:** All relevant data are within the manuscript and its Supporting Information files.

**Funding:** D.F.Q.S. and A.C. were supported in part by National Institutes of Health (https://www.nih.gov) grants R01-AI162381, R01-AI152078, and

## Abstract

Pathogenic Potential (PP) is a mathematical description of an individual microbe, virus, or parasite's ability to cause disease in a host, given the variables of inoculum, signs of disease, mortality, and in some instances, median survival time of the host. We investigated the relationship between pathogenic potential (PP) and infective inoculum (I) using two pathogenic fungi in the wax moth *Galleria mellonella* with mortality as the relevant outcome. Our analysis for *C. neoformans* infection revealed negative exponential relationship between PP and I. Plotting the log(I) versus the Fraction of animals with signs or symptoms (Fs) over median host survival time (T) revealed a linear relationship, with a slope that varied between the different fungi studied and a y-intercept corresponding to the inoculum that produced no signs of disease. The I vs Fs/T slope provided a measure of the pathogenicity of each microbial species, which we call the pathogenicity constant or $k_{Path}$. The $k_{Path}$ provides a new parameter to quantitatively compare the relative virulence and pathogenicity of microbial species for a given host. In addition, we investigated the PP and Fs/T from values found in preexisting literature. Overall, the relationship between Fs/T and PP versus inoculum varied among microbial species and extrapolation to zero signs of disease allowed the calculation of the lowest pathogenic inoculum (LPI) of a microbe. Microbes tended to fall into two groups: those with positive linear relationships between PP and Fs/T vs I, and those that had a negative exponential PP vs I relationship with a positive logarithmic Fs/T vs I relationship. The microbes with linear relationships tended to be bacteria, whereas the exponential-based relationships tended to be fungi or higher order eukaryotes. Differences in the type and sign of the PP vs I and Fs/T vs I relationships for pathogenic microbes suggest fundamental differences in host-microbe interactions leading to disease.

## Author summary

The ability of a microbe, virus, or parasite to cause disease is dependent on multiple factors, virulence factors. host immune defenses, the infective inoculum, and the type of immune response. For many microbes their capacity for causing disease is highly dependent on the inoculum. The mathematical formula for Pathogenic Potential (PP) is a way to compare the ability of an organism to have a pathogenic effect, as measured by Fraction

R01-HL059842. The funders had no role in study design, data collection and analysis, decision to publish, or preparation of the manuscript.

**Competing interests:** The authors have declared that no competing interests exist.

with signs or symptoms (Fs), mortality (M), and inoculum (I), and can include the median survival time of the host (T). Increasing inoculum of the fungus *Cryptococcus neoformans* for a moth host resulted in exponentially smaller pathogenic potential, and the Fs/T versus inoculum plot showed a logarithmic relationship. Together, these relationships show diminishing returns with increasing cryptococcal inoculum, in which each individual fungus plays a smaller role in pathogenicity. Literature data shows that other microbes, mostly bacteria, had linear Fs/T versus inoculum relationships, which indicate that each bacterium contributed an equal amount to pathogenicity. These differences in relationships can point to differences in host-microbe interactions and suggest new ways in which the organism causes disease.

## Introduction

The pathogenic potential (PP) of an organism was proposed in 2017 as an attempt to develop a quantitative method that would allow comparing the capacity for virulence of different microbial species [1] and is defined by the equation:

$$PP = \frac{Fs}{I}(10^M) \qquad \text{Eq 1}$$

whereby Fs is the fraction of the population with microbe-relevant signs or symptoms, I is the infective inoculum, and M is the mortality fraction. Mortality, as the $10^M$ term, was included as a separate variable in order to amplify the pathogenic potential of lethal microbes versus non-lethal ones. In host-microbe interactions that do not result in host death that M = 0.0 and the term $10^M$ becomes 1. Later this concept was expanded by showing how PP could be used to estimate the contribution of virulence factors to pathogenicity, and by adding the parameter of time, described as $PP_T$ (Eq 2), to account for the fulminant nature of some infectious diseases [2].

$$PP_T = \frac{Fs}{IT}(10^M) \qquad \text{Eq 2}$$

The initial equations were written assuming that the various parameters were linearly related as a first approximation, partly for simplicity and partly because there was no evidence to the contrary. However, proposing a PP equation raised the question of what the actual mathematical relationship between such parameters as Fs and I was, which in turn suggested the need for experimental measurements using pathogenic organisms in a susceptible host. A further question was whether there were differences between these parameters in different microbial species or hosts. For example, vertebrates have both innate and adaptive immune responses that neutralize microbes, whereas invertebrates have only an innate-like immune response. Further, the mechanisms by which microbes damage hosts and cause disease vary widely. Disease occurs when the host has suffered sufficient damage such that homeostasis is altered and this damage can come from direct microbial action, the immune response, or both [3].While each pathogenic microbe is different and generalizations are difficult, bacteria tend to cause disease through routes of tissue damage and toxicity, whereas fungi cause disease through growth in tissues and persistence within the host, and for both host damage results from microbial action and the immune response. Consequently, we hypothesized that differences in mechanism of disease could be reflected in differences in the relationships between the measures of pathogenicity and inoculum.

In this study we used the *Galleria mellonella* system [4] to explore the relationship between I and Fs. This system is particularly attractive because it is a non-vertebrate animal host that is highly susceptible to many pathogenic microbes. Our analysis reveals a non-linear relationship between PP and I and suggest that the slope of the relationship between I and Fs can be used for a quantitative comparison of the relative virulence of microbial species. We investigated the existing literature to evaluate whether this exponential relationship between PP and I was universal or unique to *C. neoformans* in the *G. mellonella* host, and found that other microbes, predominately bacteria, had linear PP vs I and Fs/T relationships, whereas fungi tended to have the exponential relationships seen with *C. neoformans*. Further, we see the same exponential relationships with *C. neoformans* infections of murine hosts as we do in *G. mellonella* hosts. Our results suggest that the types of mathematical relationships can differ for individual pathogenic microbes and that these differences can reveal fundamental differences in virulence strategies and/or host responses to infection.

## Results

### Pathogenic Potential for *Cryptococcus neoformans* in *Galleria mellonella*

We analyzed the pathogenic potential (PP) of *C. neoformans* H99 strain when infecting *Galleria mellonella* at an inoculum of $10^5$ cells/larvae and incubated at 30˚C from sixteen different experiments (Fig 1A) and found that the average PP was 8.64 x $10^{-5}$ (Fig 1B). We similarly calculated the $PP_T$, which is a measurement of pathogenic potential as it relates to time until death in 50% of hosts ($LC_{50}$)[2]. We found that the average $PP_T$ of *C. neoformans* at this inoculum was 1.23 x $10^{-5}$ (Fig 1B). These data from 16 independent experiments shows the variation in PP measured in one laboratory and provides a range to consider when comparing calculated PP and $PP_T$ from other organisms using literature values below.

### Correlation of PP and $PP_T$ as a Function of Inoculum

To understand the relationship between inoculum and PP and $PP_T$, we infected *G. mellonella* with *C. neoformans* using different inoculums (Fig 2A). We observed that as inoculum

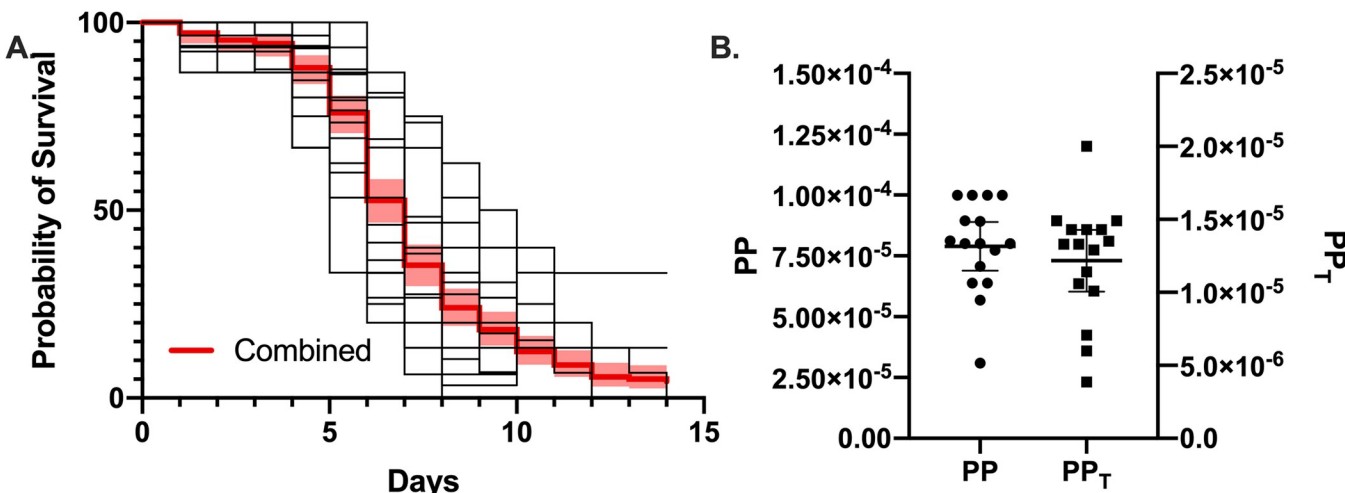

**Fig 1. *Pathogenic Potential of* C. neoformans *in* G. mellonella.** (A). Overlapping plots of the survival of *G. mellonella* infected with *C. neoformans* at an inoculum of $10^5$ cells/larvae. Each of the 16 survival curves represents a replicate infection with 15 to 30 larvae. The red line indicates the combined survival curve with a 95% confidence interval. The individual pathogenic potential (PP) (B) and pathogenic potential in respect to time ($PP_T$) were calculated and plotted. Each data point in (B) represents the calculated PP or $PP_T$ of an individual experiment. Error bars represent mean with 95% confidence interval.

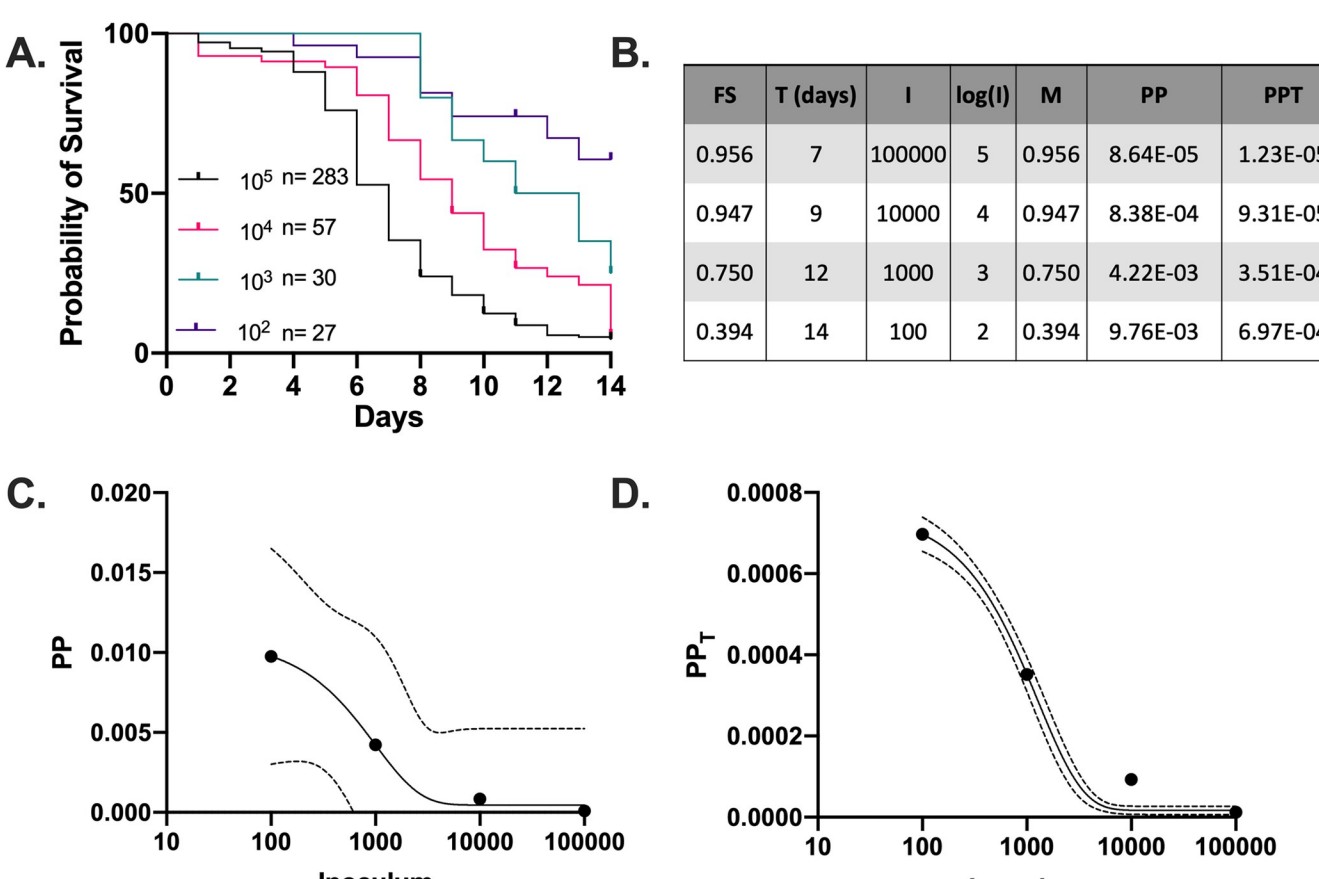

**Fig 2. *Pathogenic Potential of* C. neoformans *as a function of inoculum*.** (A) Survival curves of *G. mellonella* infected with different inocula of *C. neoformans*, and the calculated Fs, T, M, and pathogenic potentials (PP and $PP_T$) (B). Plots of PP (C) and $PP_T$ (D) versus I on log-scaled x-axes. These show a negative exponential relationship between pathogenic potential and inoculum, as fitted by a one phase exponential decay function. 95% CI of the exponential fit line is shown with dotted lines.

increased, there was an expected decrease in time to death until 50% of host organisms died, with an increase in Fraction with signs or symptoms (Fs) and Mortality (M) (Fig 2B). We also observed a negative exponential decrease in PP and $PP_T$ while inoculum increases (Fig 2C and 2D). In both measures, the lower inoculum was associated with a higher pathogenic potential. The negative exponential relationship between pathogenic potential and inoculum implies that the average microbe during an infection with high inoculum makes smaller contribution to the outcome of infection than in a lower inoculum infection.

## Correlation of Fs/T as a Function of Inoculum

Plotting Fs versus Inoculum yielded logarithmic curves (Fig 3A). Similarly, a plot of Fs/T versus I revealed a logarithmic relationship (Fig 3B). The higher the inoculum, the higher the Fs and Fs/T values are. Further, the relationship between Fs/T and the log of the inoculum was linear, indicating a direct correlation between log(I) and Fs/T (Fig 3C), implying that a simple line equation described that relationship. Since this relationship between inoculum and disease is logarithmic and not linear, it implies that microbes the higher inoculum on average contribute less to the outcome of infection. This would be consistent with a microbe that causes disease from a high microbial burden in the host due to exponential doubling of microbes. From

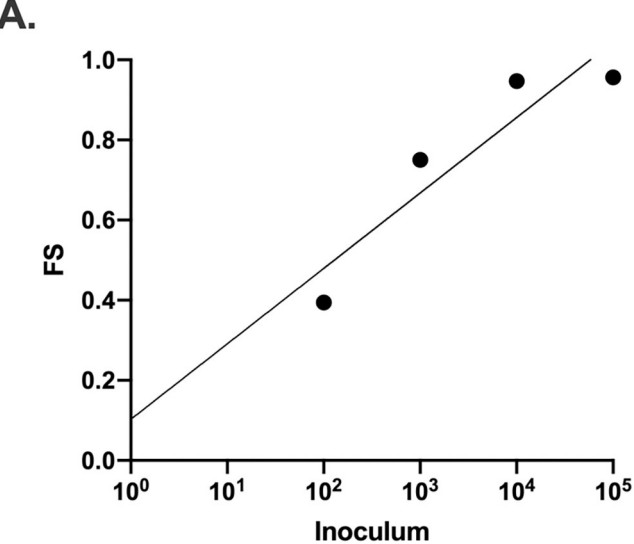

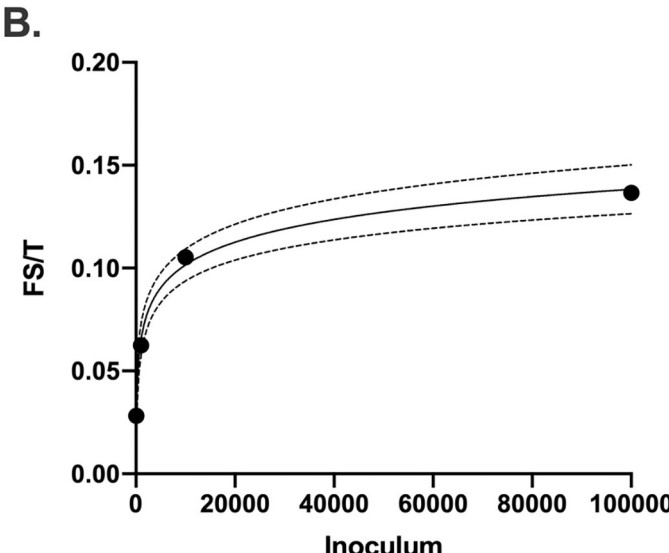

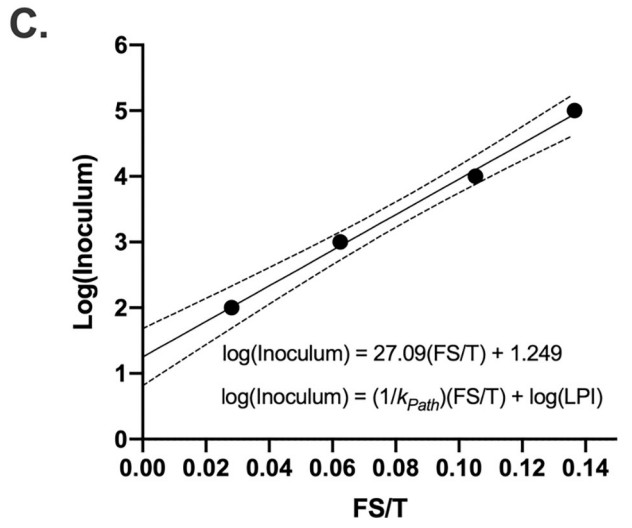

**Fig 3.** *Determination of* $k_{Path}$ *for* **C. neoformans** *in G. mellonella.* (A) Fraction with signs or symptoms and (B) Fraction with signs or symptoms relative to the $LT_{50}$ for larvae infected with different inocula of *C. neoformans* plotted on a log-scaled x-axis. These show that there is a positive logarithmic relationship between Fs and Fs/T versus inoculum, as fitted by a semi-log line, in which the x-axis is logarithmic, with 95% CI shown as dotted lines (A and B), or a simple linear regression for log(I) vs. Fs/T (C). This relationship can be used to calculate the pathogenicity constant ($k_{Path}$) and the lowest pathogenic inoculum (LPI) (C).

this line equation, we could derive the y-intercept, which would be the smallest inoculum to cause a pathogenic effect with regards to time (Fs/T), which we termed the Lowest Pathogenic Inoculum (LPI) (Fig 3C). Similarly, the slope provided information on how initial inoculum is related to the outcome of the host, and by virtue of being a slope is a constant value that describes the microbe's pathogenic nature regardless of inoculum.

## Similarities between PP, PPT, and Fs/T in Mice and *G. mellonella* models

Calculating PP and $PP_T$ for H99 murine infections showed similar trends as the *G. mellonella* data (Fig 4A and 4B), with both having negative exponential relationships between the measures of pathogenicity and the inoculum of infection, for the different mouse strains and route of infection. This suggests similar relationships between the host and *C. neoformans* in both *G. mellonella* and murine models. When calculating Fs/T values from H99 murine infections, we found similar trends in the Fs/T values, indicating similar relationships between the host and *C. neoformans* in both *G. mellonella* and murine models (Fig 4C and 4D). The data also

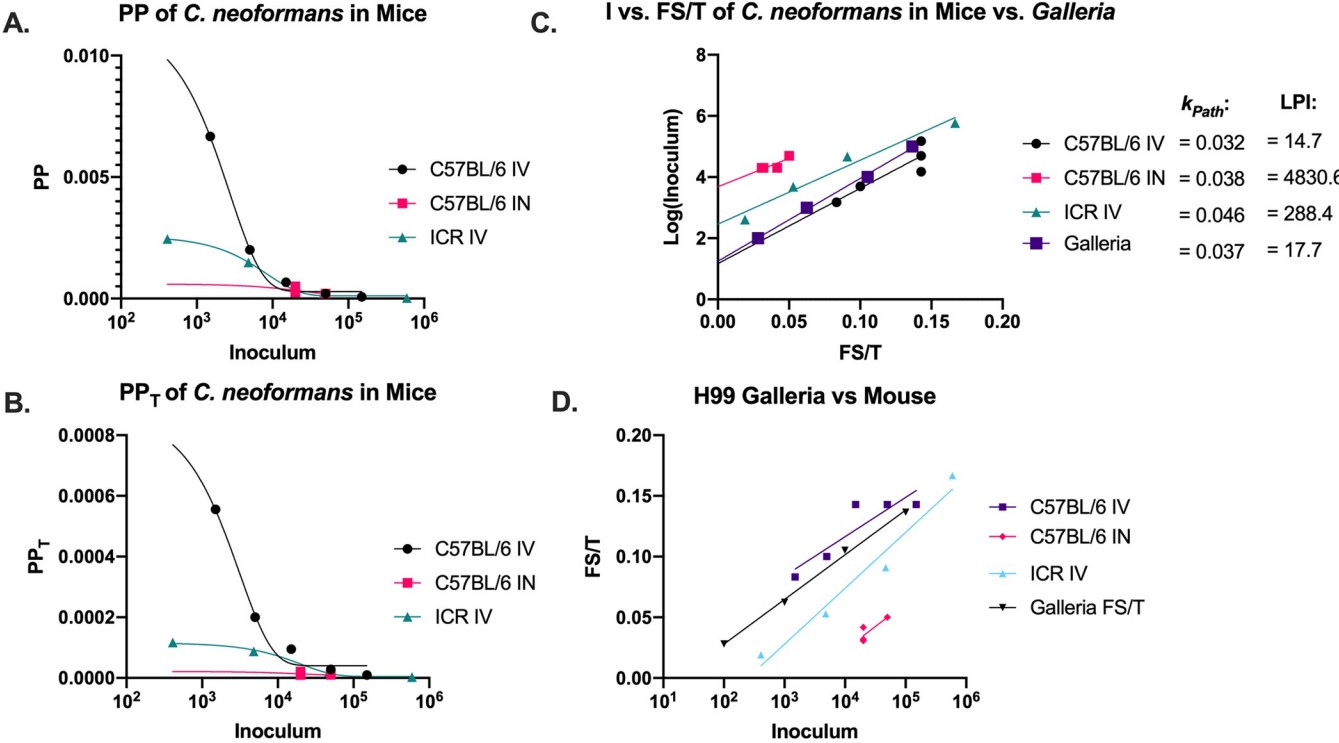

**Fig 4.** *Pathogenic Potential of* **C. neoformans** *in mice.* Using literature values [47–49], we calculated the (A) pathogenic potential (PP), (B) pathogenic potential in respect to time ($PP_T$), (C) Fs/T, (C) lowest pathogenic inoculum (LPI), and (C) $k_{Path}$ for *C. neoformans* in mouse models through various inoculation routes. Generally, the trends were consistent between the fungus in *G. mellonella* and murine hosts. (A) PP vs I and (B) $PP_T$ vs I data was fitted by a one phase exponential decay function, (C) log(I) vs. Fs/T was fitted by a linear regression, and (D) Fs/T vs I data was fitted by a semi-log line in which the x-axis is logarithmic. The (C) log(I) vs. Fs/T and (D) Fs/T vs I slopes were similar between the two hosts, indicating similar $k_{Path}$ values.

indicated lowest pathogenic inoculums (LPI) that varied by mouse strain and route of infection, some of which were comparable to the LPI of *C. neoformans* in *G. mellonella*. For intravenously infected C57BL/6 mice, the LPI was 14.7 cells while for intranasal infection the LPI was 4830 cells. The intravenous-infected C57BL/6 also had a lower LPI than the intravenous-infected ICR strain (288 cells), which could be indicative of immune variations between the strains.

## Pathogenicity Constant for *C. neoformans*

We observed a linear relationship between Fs/T and log(I) and noted that the slope of this linear best-fit equation incorporated all the components of pathogenicity (i.e fraction with signs of disease, median time until death ($LT_{50}$), and inoculum into a value that is constant at all inoculums. This constant value (slope) could allow comparisons between microbial strains and species even between experiments performed at different inoculum, which is where the PP and $PP_T$ values have their limitations. Using the equation of the line derived from Fig 3C, this pathogenicity constant, $k_{Path}$, can be described by (Eq 3.1–3.2).

$$\log(I) = \left(\frac{1}{k_{Path}}\right)\frac{Fs}{T} + \log(LPI) \qquad\qquad \text{Eq 3.1}$$

$$k_{Path} = \frac{Fs}{T[\log(I) - \log(LPI)]} \qquad\qquad \text{Eq 3.2}$$

The calculated value of $k_{Path}$ for *C. neoformans* (H99) infection of *G. mellonella* is 0.0369 based on our experimental data. We calculated a $k_{Path}$ for *C. neoformans* infections in mice ranging from 0.032 to 0.046 depending on the mouse strain, route of infection, and study (Fig 4C), which is comparable in magnitude to that for *G. mellonella*. The $k_{Path}$ value is defined as the fraction of hosts with signs of disease per $LT_{50}$ log inoculum. Essentially, $k_{Path}$ is a measure of how fast the hosts get sick and die per log inoculum. High values represent microbes that cause greater and faster damage with each additional order of magnitude of cells, conversely, smaller values represent microbes that cause a steady, slower pathogenicity in which additional orders of magnitude of cells do not have a substantial effect.

## Fungal PP, $PP_T$, Fs/T and $k_{Path}$ in *G. mellonella*

From these insights with the *C. neoformans-G. mellonella* system we explored their applicability to other pathogenic microbes and analyzed published *G. mellonella* data to calculate the experimental PP, $PP_T$, Fs/T and $k_{Path}$ of other fungi. For the entomopathogenic fungus *Beauveria bassiana*, the relationships between fungal inoculum and PP, $PP_T$, and Fs/T were each similar to those calculated for *C. neoformans* with a slightly higher $k_{Path}$ equal to 0.1 (Fig 5A, 5B and 5C) [5,6]. However, we saw different trends for the three other fungal species. In the case of *Candida albicans*, there was no clear relationship between inoculum and PP and $PP_T$, however, the Fs/T versus I relationship was logarithmic, like *B. bassiana* and *C. neoformans*, but with a much steeper slope, and thus the higher $k_{Path}$ of 0.566. (Fig 5D, 5E and 5F, black) [7,8]. Similar trends and values were seen in *G. mellonella* infections performed by our group (Fig 5D, 5E and 5F, teal). The steeper $k_{Path}$ and the higher LPI indicate there is a higher barrier for the fungus to be pathogenic, but once that threshold is met, pathogenicity increases rapidly. For *G. mellonella* infected with *Histoplasma capsulatum* and *Paracoccidioides lutzii*, the plotting yielded negative exponential relationships between inoculum and PP and $PP_T$, and an Fs/T vs I relationship that was essentially flat with a $k_{Path}$ value near zero (Fig 5G, 5H and 5I) [9]. Essentially, based on the Fs/T vs I and $k_{Path}$ values, there was no inoculum-dependent

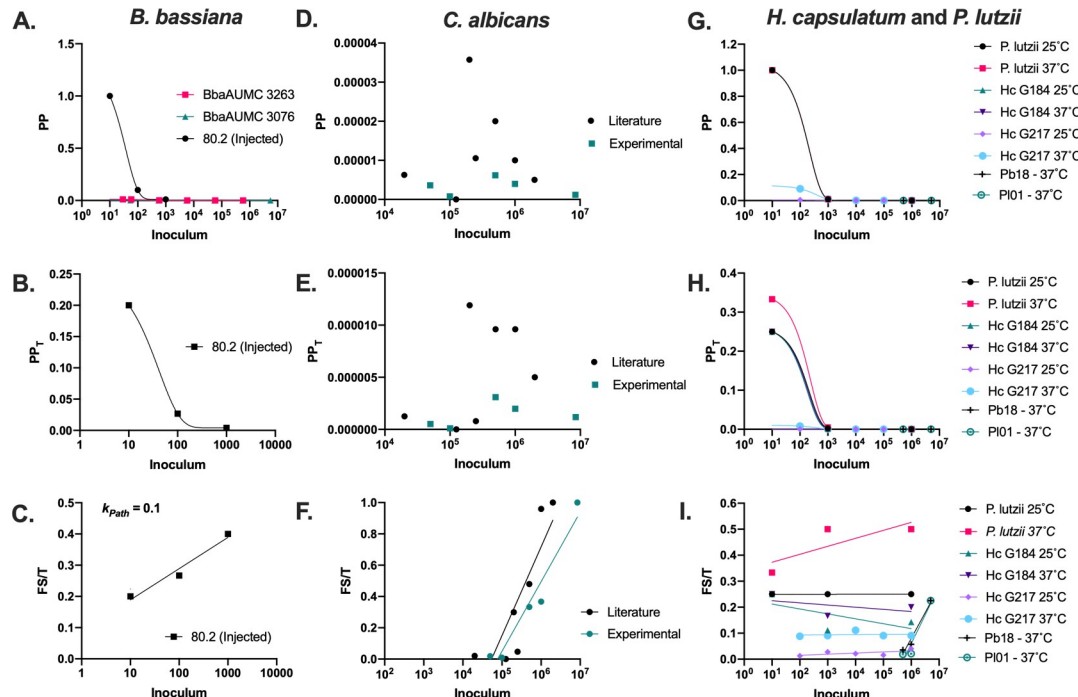

**Fig 5. _Pathogenic potentials of other fungi in_ G. mellonella _hosts_.** Using existing published values *[5–10]*, we calculated (A) PP, (B) PP_T, and (C) Fs/T for the entomopathogenic fungus *Beauveria bassiana's*. These showed similar relationship to inoculum as *C. neoformans*. Similarly, we calculated *C. albicans'* (D) PP, (E) PP_T, (F) and Fs/T and plotted it versus inoculum from previously published and new experimental data. We did not see a clear association of PP and PP_T with the inoculum, however, there was a logarithmic relationship between the inoculum and Fs/T (F). For *Histoplasma capsulatum*, *Paracoccidioides lutzii*, and *Paracoccidioides brasiliensis*, we used literature sources to calculate the (G) PP, (H) PP_T, and (I) Fs/T vs. inoculum with different strains and temperatures and found that the PP and PP_T mostly had a relationship with inoculum that was best fitted by a one phase exponential decay line. The Fs/T values were mostly independent of inoculum used, with the exception of the Pb18 and Pl01 strains at higher inoculums. (A,D,E) PP vs I and (B,E,H) PP_T vs I data was fitted by a one phase exponential decay function, and (C,F,I) Fs/T vs data was fitted by a semi-log line in which the x-axis is logarithmic.

mortality for the infected larvae for these two pathogenic fungi. However, in one study [10] that used a higher inoculum, there was a dose-dependent effect on host death, where larvae infected with 5 x 10^6 cells died faster than those infected with 1 x 10^6. Future studies may want to further investigate the mechanism underlying the unique dose-dependency, or independency, of *H. capsulatum* and *P. lutzii* infections in *G. mellonella*. Further investigation may include quantification of the reported dose-dependent melanization response in larvae, which could be used as the Fs value and provide more nuanced and intuitive inoculum dose-dependency in PP, PP_T, and Fs/T. The general dose-independent effect on survival could be the result of the slow and irregular growth of the microbe [11–13], or a damaging immune response that kills the host in response to few or many microbes (Table 1). In this regard, *P. lutzii*, *P. brasiliensis*, and *H. capsulatum* are both slow growing fungi with doubling rates in media ranging from 13 to 21 hours [11–13], compared with the ~2 hour doubling time of *C. neoformans* in culture [14] and ~5 hours *in vivo* during infection of *G. mellonella* hosts [15]. Associations between *P. brasiliensis* growth rate and virulence have been previously indicated [16]. Additionally, the higher temperatures for the 37°C conditions used in these experiments is a variable that may cause thermal stress on the larvae that could impact their immune response and baseline longevity compared to the 25°C incubation condition [17,18]. Dissimilar to the findings in *G. mellonella*, analysis of murine infection with *P. brasiliensis* reveals an

**Table 1.** *Relationships between PP, PP$_T$, and Fs/T with inoculum, proposed explanation, and examples of microbes.*

| Relationships | Explanation | Examples |
|---|---|---|
| **PP vs I is Positive Linear/Exponential** | Each microorganism contributes a measurable amount of pathogenicity directly. Disease is possibly mediated by a toxin or compound produced by the organism. | *S. aureus, S. agalactiae* |
| **PP$_T$ vs I is Positive Linear/Exponential** | Each microorganism contributes a measurable amount of pathogenicity including time to death. Disease is possibly mediated by a toxin or compound produced by the organism. | *S. aureus, S. agalactiae, P. aeuriginosa* |
| **Fs/T vs I is Positive Linear** | Speed of disease onset and death is directly related to number of microorganisms present in the infective inoculum. Possibly indicates that time until death mediated by a toxin or compound produced by the organism. | *L. monocytogenes, S. aureus, S. agalactiae, P. aeuriginosa* |
| **Fs/T vs I is Positive Logarithmic** | Speed of disease onset is related to the number of microorganisms present in the infective inoculum. Thus, additional organisms have less individual impact on speed of disease. Disease is possibly mediated by organisms' ability to grow and their doubling time. | *C. neoformans, C. albicans, B. bassiana* |
| **PP vs I is Negative Exponential** | Pathogenicity is related to the number the microorganisms in infective inoculum. Thus, additional organisms have less individual impact on pathogenicity. This indicates the disease is possibly mediated by organisms' ability to grow. | *C. neoformans, L. monocytogenes, B. bassiana, GmNPV* |
| **PP$_T$ vs I is Negative Exponential** | Pathogenicity over time is related to the number of microorganisms present in the infective inoculum. Additional organisms have less individual impact on pathogenicity. Disease is possibly mediated by organisms' ability to grow and their doubling time. | *C. neoformans, L. monocytogenes* |
| **Fs/T vs I is Flat** | Speed of disease progression and mortality is not dependent on number of organisms. Such curves potentially due to slow growth, host immune response, or toxicity. | *H. capsulatum, P. brasiliensis* |

inoculum-dependency for PP, PP$_T$, and Fs/T similar to what is seen with other fungi (S1A, S1B, S1C, and S1D Fig) [19]. There is no clear inoculum-dependent effect on PP or PP$_T$ in *H. capsulatum* infection of mice, and there is a roughly positive linear relationship between Fs/T versus I (S1E, S1F, S1G and S1H Fig) [20,21], which is unlike other fungal Fs/T vs. I relationships observed in *G. mellonella*.

## Bacterial PP, PP$_T$, Fs/T in *G. mellonella*

Next, we considered data found in literature that would allow us to calculate PP, PP$_T$ and Fs/T for bacterial infections of *G. mellonella* [22–27]. In general, the relationships between pathogenicity and inoculum for bacteria were different from those relationships in fungi. For example, all the bacterial species analyzed, aside from *Salmonella enterica* Typhimurium had an Fs/T vs I relationship that was linear, compared to the logarithmic one in fungi (Fig 6C, 6F, 6I, 6L and 6O. This indicates a direct relationship between disease progression over time and the starting inoculum, rather than one related to the inoculum's order of magnitude (log[I]). The positive linear relationship between Fs/T and inoculum indicates that microbes contribute equally to disease during low and high inoculum infections, meaning that each bacterium makes a set contribution to disease. This is expected with microbes that produce of toxins or inflammatory molecules that work in a dose dependent manner. Because of this, the $k_{Path}$ formula described above would not be accurate for *Salmonella*, however, it could be modified to simply be a metric like the PP$_T$ value without the consideration of mortality:

$$k_{Path} = \frac{Fs}{T[(I) - (LPI)]} \qquad \text{Eq 4}$$

There was also variation between the PP vs I and PP$_T$ vs I relationships in bacteria, where the relationships were positive and linear, as opposed to the negative exponential ones in the fungi we analyzed (Fig 6A, 6B, 6D, 6E, 6F, 6G, 6H, 6J, 6K, 6M and 6N). This would suggest that in infections of these species (*S. aureus.*, *P. aeruginosa*, and *Streptococcus spp.*) that each additional bacterium causes a set unit of damage, whereas for fungi, there are diminishing

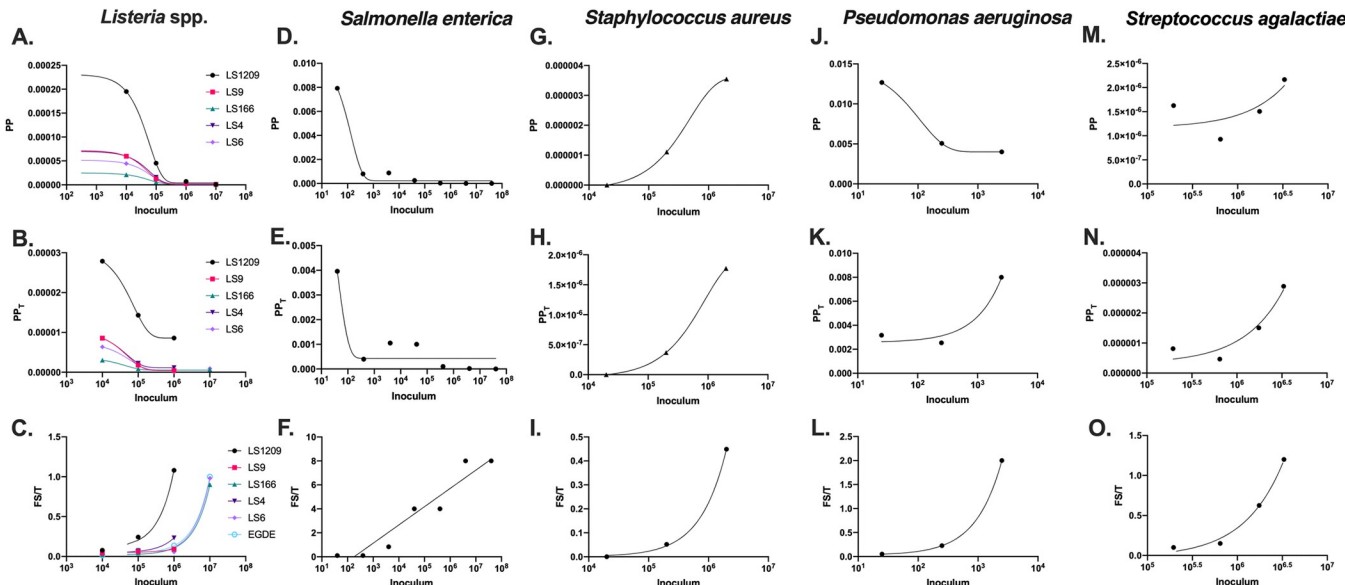

**Fig 6. *Pathogenic potentials of bacterial species in* G. mellonella *hosts.*** Using literature values *[22–27]* we calculated the Pathogenic Potential (PP), Pathogenic Potential in regards to time ($PP_T$), and Fs/T for (A-C) *Listeria spp.*, (D-F) *Salmonella enterica*, (G-I) *Staphylococcus aureus*, (J-L) *Pseudomonas aeruginosa*, and (M-O) Group B *Streptococcus*. Overall, we found various relationships between the measures of pathogenic potential and the bacterial inoculum that varied species to species. While most of the (A, D, J) PP values had a negative exponential relationship with the inoculum and are best-fitted with an exponential decay function, *S. aureus* had positive exponential relationships between the (G) PP and (H) $PP_T$ versus inoculum, and (M,N) *Streptococcus* and (K) *P. aeruginosa* ($PP_T$ only) had positive linear relationships between the PP and $PP_T$ versus inoculum, best-fitted with a simple linear regression. All the bacterial species investigated besides (F) *Salmonella enterica* had a linear Fs/T vs. I relationship, which is inconsistent with what is seen in fungi. The linear relationship indicates each bacterium influences the degree and speed of death, rather than the order of magnitude of bacteria.

returns with increasing inoculum with regards to damage from each additional fungal cell. There does not seem to be an association between the positive linear PP, $PP_T$, and Fs/T relationships and whether the bacteria are Gram-negative or Gram-positive. However, this pattern would suggest there is a dose-dependent effect causing death in the *G. mellonella* larvae, such as the secretion or production of a toxin or inflammatory molecule (Table 1).

## PP, $PP_T$, and Fs/T of entomopathogenic nematodes in *G. mellonella*

*G. mellonella* are common models for infection with entomopathogenic nematodes, including the purpose of culturing the nematodes and even using them as bait to collect nematode species in the wild. We calculated the PP, $PP_T$, and Fs/T for two entomopathogenic nematode species [28] in *G. mellonella*. The PP and $PP_T$ vs I relationships, like those seen in *C. neoformans*, *L. monocytogenes*, and *Salmonella enterica*, manifested a negative exponential trend, with some variability in the middle inoculum infections (Fig 7A, 7B, 7D and 7E). The Fs/T vs I curve was positive and roughly linear, although it has a sigmoidal shape, closely fitted by an exponential one phase decay line (Fig 7C and 7F). It is worth noting these nematodes themselves do not kill the insect larvae. Once the larvae are infected with the nematodes, the nematodes release bacteria that are highly pathogenic and encode toxins that kill the host.

## PP of the *G. mellonella* Nuclear Polyhedrosis Virus (*Gm*NPV)

We calculated the PP of the *G. mellonella* Nuclear Polyhedrosis Virus (*Gm*NPV), which is a baculovirus that primarily infects Lepidoptera. The results of Stairs' 1965 study [29], yielded a clear negative exponential relationship between PP and inoculum of virus, whereas the data

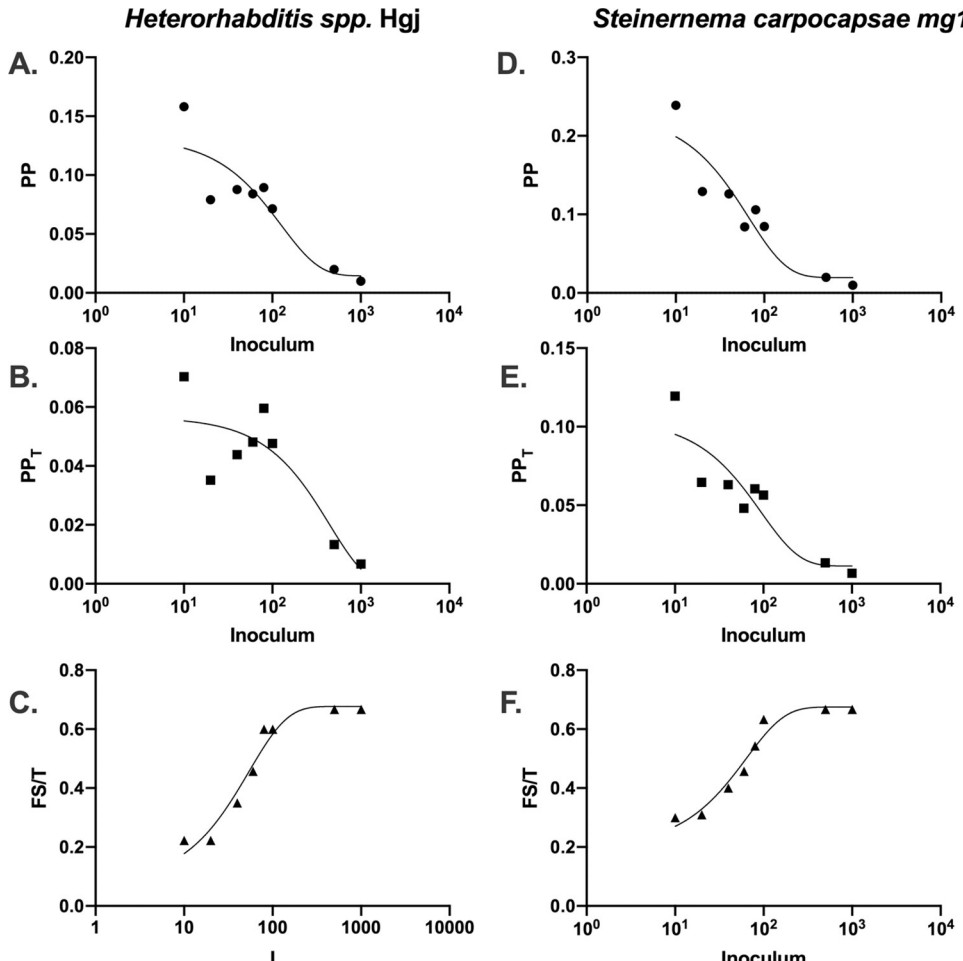

**Fig 7.** *Pathogenic Potential of Nematodes in* **G. mellonella** *hosts.* Using literature values [28], we calculated the PP, $PP_T$, and Fs/T for the entompathogenic nematodes (A-C) *Heterorhabditis spp*. strain Hgj and (D-F) *Steinernema carpocapsae* strain mg1. Generally, there were exponential PP vs. I and $PP_T$ vs. I relationships (as fitted by a one phase exponential decay function), as seen with fungi and some bacteria, with some variation in the middle-inoculum groups. The (C,F) Fs/T vs I relationships were best fitted by a one phase exponential decay (exponential plateau) function.

from Fraser and Stairs' 1982 study [30] yielded an inverted U-shaped curve with an exponential negative relationship at the higher viral inoculum (Fig 8).

## Modeling relationships between pathogenicity and inoculum

After noting various relationships between the pathogenicity metrics (PP, $PP_T$, Fs/T) and inoculum we sought to understand how these differences occurred. Hence, we modeled PP, $PP_T$, and Fs/T calculations for a hypothetical microbe at different inoculum (Fig 9). For one microbe, we calculated the Fs value as a direct function of the inoculum, represented by Eq 5 and 6, where $x_1$ and $y_1$ represent variables dependent on the mortality, Fs, T, and I of the infection (Eq 5.1 and 6.1). For the purposes of Fig 9, we used x = $10^{-5}$ and y = $10^5$.

$$Fs = x_1 \times I \qquad\qquad \text{Eq 5}$$

$$x_1 = Fs/I \qquad\qquad \text{Eq 5.1}$$

## *Gm*NPV PP vs I

**Fig 8. Gm*NPV Pathogenic Potential.*** The pathogenic potential of the *Gm*NPV (nuclear polyhedrosis virus) was calculated from published values [29,30] and plotted against inoculum. There is a negative exponential relationship between the amount of virus used to infect *G. mellonella* and the pathogenic potential in the Stairs 1965 study. In the Fraser and Stairs 1982 study, the relationship is varied, where the lower inocula have a positive exponential relationship with pathogenic potential, and the higher inocula have a negative exponential relationship with PP. Both plots are fitted with an exponential one phase decay function.

$$T = \frac{y_1}{I} \qquad \text{Eq 6}$$

$$y_1 = IT \qquad \text{Eq 6.1}$$

Plotting the PP, $PP_T$, and Fs/T values revealed a pattern similar as expected (Fig 9, black data points). For the second microbe, we aimed to model disease progression based on the magnitude of the inoculum, and in doing so, used Eq 7 and 8, where $x_2$ and $y_2$ represent variables dependent on the mortality, Fs, T, and log(I) (Eqs 7.1 and 8.1). For the purposes of Fig 9, we used x = 0.1 and y = 10.

$$Fs = x_2 \times \log(I) \qquad \text{Eq 7}$$

$$x_2 = Fs/\log(I) \qquad \text{Eq 7.1}$$

$$T = \frac{y_2}{\log(I)} \qquad \text{Eq 8}$$

$$y_2 = T\log(I) \qquad \text{Eq 8.1}$$

This resulted in PP, $PP_T$, and Fs/T values that when plotted yielded negative exponential PP and $PP_T$ and a positive logarithmic Fs/T, such as *C. neoformans* and *B. bassiana* (Fig 9, pink data points).

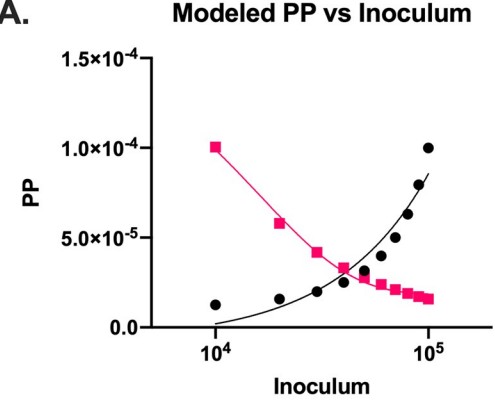

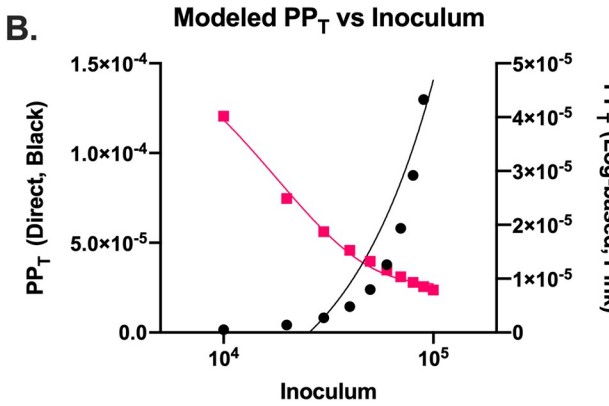

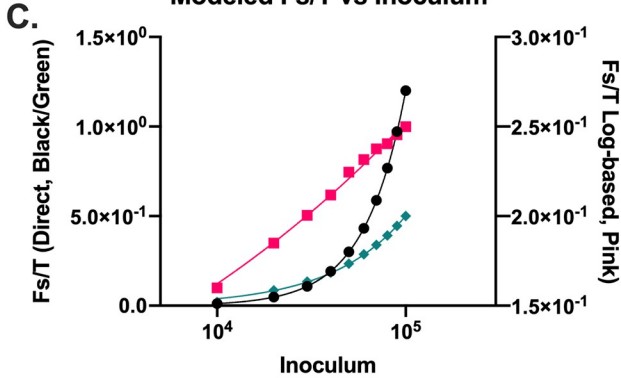

**Fig 9. Modeled PP, PPT, and Fs/T values. Modeled PP** (A), $PP_T$ (B), and Fs/T (C) values using linear-based methods of calculating Fs and T (black data points) or log-based methods of calculating Fs and T (pink data points), or a mix of both (green data points), as described by the formulas in the graph key. Example organisms that fall under each category are listed below their respective group. PP and $PP_T$ values are fitted with a one phase exponential decay function. The linear based Fs/T values (black and green points) are fitted using a simple linear regression, whereas the log-based values (pink points) are fitted using a semi-log line.

## Calculating PP, PP$_T$, and Fs/T across microbes for the same infectious inoculum

Through the fitted exponential or linear lines for the PP, PP$_T$, and Fs/T versus I plots, we are able to use the equations of the line to calculate theoretical PP, PP$_T$, and Fs/T values for infectious inoculums that have not yet been experimentally studied. This provided a way to compare measures of pathogenicity amongst microbes, even when the original experiments are performed at different inoculum. These calculated values are found in Table 2. It is worth noting that these values are preliminary, and based on literature, and should not be taken as definitive until experimentally confirmed using the exact inoculum. It can, however, be used to approximate disease severity outcomes when planning experimental design.

**Table 2.** *Calculated PP, PP$_T$, Fs/T, and k$_{Path}$ values for inoculum of 10$^5$ organisms or virions.* tested.

| Organism | PP[a] | PP$_T$[a] | Fs/T[a] | $k_{Path}$ | Reference |
|---|---|---|---|---|---|
| *Cryptococcus neoformans* | $4.60 \times 10^{-4}$ | $1.70 \times 10^{-5}$ | $1.38 \times 10^{-1}$ | $3.69 \times 10^{-2}$ | This Work |
| *Candida albicans* | N/A | N/A | $1.51 \times 10^{-1}$ | $5.67 \times 10^{-1}$ | [7,8] |
| *Beauveria bassiana* | | | | | |
| 80.2 (Injected) | $1.00 \times 10^{-2}$ | $4.00 \times 10^{-3}$ | $5.89 \times 10^{-1}$ | $1.00 \times 10^{-1}$ | [6] |
| BbaAUMC 3263 | $3.20 \times 10^{-4}$ | N/A | N/A | N/A | [5] |
| BbaAUMC 3076 | $3.61 \times 10^{-5}$ | N/A | N/A | N/A | [5] |
| *Histoplasma capsulatum* | | | | | |
| G184 25°C | $1.00 \times 10^{-5}$ | $1.40 \times 10^{-6}$ | $1.37 \times 10^{-1}$ | $-1.89 \times 10^{-2}$ | [9] |
| G184 37°C | $1.00 \times 10^{-5}$ | $2.00 \times 10^{-6}$ | $1.92 \times 10^{-1}$ | $-8.33 \times 10^{-3}$ | [9] |
| G217 25°C | $5.00 \times 10^{-5}$ | $2.30 \times 10^{-6}$ | $2.79 \times 10^{-2}$ | $4.30 \times 10^{-3}$ | [9] |
| G217 37°C | $3.70 \times 10^{-4}$ | $4.00 \times 10^{-5}$ | $9.50 \times 10^{-2}$ | $5.68 \times 10^{-4}$ | [9] |
| *Paracoccidioides lutzii* | | | | | |
| Pl01 25°C | $1.00 \times 10^{-5}$ | $2.50 \times 10^{-6}$ | $2.50 \times 10^{-1}$ | $2.09 \times 10^{-17}$ | [9] |
| Pl01 37°C | $1.00 \times 10^{-5}$ | $5.00 \times 10^{-6}$ | $4.95 \times 10^{-1}$ | $3.07 \times 10^{-2}$ | [9] |
| Pl01 37°C | N/A | N/A | $-1.62 \times 10^{-1}$ | $2.21 \times 10^{-1}$ | [10] |
| *Paracoccidioides brasiliensis* | $7.79 \times 10^{-7}$ | $1.02 \times 10^{-7}$ | $-1.19 \times 10^{-1}$ | $1.98 \times 10^{-1}$ | [10] |
| *Listeria monocytogenes* | | | | | |
| LS1209 | $4.53 \times 10^{-5}$ | $1.43 \times 10^{-5}$ | $2.03 \times 10^{-1}$ | $9.78 \times 10^{-7}$ | [23] |
| LS9 | $1.27 \times 10^{-5}$ | $1.81 \times 10^{-6}$ | $5.51 \times 10^{-2}$ | $4.29 \times 10^{-8}$ | [23] |
| LS166 | $5.98 \times 10^{-6}$ | $8.55 \times 10^{-7}$ | $2.80 \times 10^{-2}$ | $8.79 \times 10^{-8}$ | [23] |
| LS4 | $1.58 \times 10^{-5}$ | $2.26 \times 10^{-6}$ | $6.53 \times 10^{-2}$ | $1.87 \times 10^{-7}$ | [23] |
| LS6 | $1.27 \times 10^{-5}$ | $1.81 \times 10^{-6}$ | $2.84 \times 10^{-2}$ | $9.56 \times 10^{-8}$ | [23] |
| EGDE | $1.32 \times 10^{-5}$ | $1.87 \times 10^{-6}$ | $4.79 \times 10^{-2}$ | $9.62 \times 10^{-8}$ | [24] |
| *Salmonella enterica* | $2.31 \times 10^{-4}$ | N/A | 4.19 | 1.52 | [25] |
| *Staphylococcus aureus* | $1.74 \times 10^{-7}$ | $5.43 \times 10^{-7}$ | $2.33 \times 10^{-2}$ | $2.24 \times 10^{-7}$ | [26] |
| *Pseudomonas aeruginosa* | $4.00 \times 10^{-3}$ | N/A | 78.9 | $7.88 \times 10^{-4}$ | [27] |
| *Streptococcus agalactiae* | $1.19 \times 10^{-6}$ | $3.95 \times 10^{-7}$ | $1.26 \times 10^{-2}$ | $3.69 \times 10^{-7}$ | [22] |
| *Heterorhabditus spp.* | $1.44 \times 10^{-2}$ | $-9.92 \times 10^{-4}$ | 1.26 | $2.54 \times 10^{-1}$ | [28] |
| *Steinernema carpocapsae* | $1.96 \times 10^{-2}$ | $1.13 \times 10^{-2}$ | 1.15 | $2.11 \times 10^{-1}$ | [28] |
| *GmNPV* | | | | | |
| | $1.34 \times 10^{-5}$ | N/A | N/A | N/A | [30] |
| | $1.34 \times 10^{-5}$ | N/A | N/A | N/A | [29] |

[a] Value calculated using 10$^5$ organisms or virions as the inoculum.

## Discussion

The concept of Pathogenic Potential (PP) was spawned from the notion that all microbes have some capacity to cause disease if acquired by a host in sufficient numbers. Disease occurs when the host has incurred sufficient damage to affect homeostasis and host damage can come from direct microbial action (e.g., toxins), the host immune response, or both (Casadevall & Pirofski 1999). According to this view, no microbes can be unambiguously labelled as either pathogens or non-pathogens, since pathogenicity is dependent on inoculum, host immunity, and other factors that affect the outcome of the host-microbe interaction [1]. In this work, we experimentally derived values for the PP and $PP_T$ for the fungi *Cryptococcus neoformans* in the invertebrate model organism *Galleria mellonella* and analyzed literature data with our mathematical equations. This analysis revealed deep differences between pathogenic microbes that are interpreted as reflecting different type of virulence mechanisms. To place this work in the context of discovery, we rely on the process of "seeking new laws" proposed by Richard Feynman for how of the laws of nature are identified [31]. Previous papers have imagined the concept of Pathogenic Potential [1,2], or as Feynman would say, these works have "guess[ed] it," which he describes as the first step in seeking new laws to describe the natural world [31]. In this work, we undertook the next step, which according to Feynman, is to, "compute the consequences of the guess," or in other words, to experimentally determine the guess' validity, then further expand the comparisons to additional "real-world" experiential observations. Following the insight of Feynman on the discovery of natural laws, this work can be considered the next step whereby the experimental work is done to to confirm or disprove the yet-to-be established theoretical equations. The current data supports the insight that microbes have diverse relationships between Pathogenic Potential and inoculum.

### Insights for Pathogenic Potential versus fungal inoculum

For *C. neoformans*, we investigated how the PP and $PP_T$ correlated with the infective inoculum moth larvae. We found that infections with smaller inocula had a larger PP and $PP_T$, despite fewer host deaths (Fs and M values) and longer survival times (T). Further, this relationship was exponential, meaning that the PP and $PP_T$ values increased exponentially with decreasing inoculum. While this result may seem counterintuitive because lower inoculum would be expected to produce less severe disease in infected larvae, it makes sense when considering the survival data. For example, almost 40% of the larvae infected with $10^3$ cells of *C. neoformans* died, while less than twice as many (~75%) died from the larvae infected with ten times as many cells ($10^4$). Thus, the average fungal cell in a lower inoculum infection contributes more towards death than fungal cells in a higher inoculum infection. This relationship may be exponential because in many microbes, proliferation and growth are exponential, as evident by the doubling of yeast cells during reproduction. Although immune defenses could reduce the growth rate *in vivo*, microbial survivors would still grow exponentially albeit at lower replication rates. If the pathogenicity of a microbe is related to microbial burden within tissues, then it makes sense that the relationship between signs and symptoms, mortality, and pathogenicity and the initial inoculation concentration are also exponential relationships rather than simple linear ones.

For the purposes of this work, we calculated the Fs value using mortality of the larvae due to the consistency of mortality being reported in literature reports and the fact that mortality is an easily measured outcome of infection. We note that *G. mellonella* can exhibit other signs and symptoms of infection, including systemic melanization and reduction in movement, which could be used to calculate a Fs value independent of morality. These values are occasionally reported, but more widespread reporting of signs and symptoms of infection could be

helpful in providing more nuanced calculations in the future. In applying these concepts, Fs should be defined by what is most appropriate for the microbe and host depending on the measurable outcomes of the specific host-microbe interaction.

The relationship between Fs/T and inoculum, For *C. neoformans* infections the experimental data for the relationship between Fs/T and inoculum was logarithmic. Unlike the relationship between $PP_T$ and inoculum, the Fs/T value increased with increasing inoculum but plateaued as inoculum increased. This makes intuitive sense since the value of Fs/T roughly equates to the number of individuals with signs of disease or deaths over time. Plotting the linear relationships of Fs vs. inoculum and Fs/T vs. inoculum allowed us to derive the minimum inoculum required to cause disease and death. These relationships for *C. neoformans* infection in *G. mellonella* larvae were generally conserved in mammalian models of infection using different mouse backgrounds and through different inoculation routes. Our calculated LPI for *C. neoformans* was one order of magnitude lower for intravenous infection than intranasal infection, which may be reflective of the extra physical and immunological barrier of the respiratory mucosa. The consistency of results between mice and moths suggests that *C. neoformans* causes disease in a similar manner in both hosts, and that the resulting relationships are due to a property of the fungus and/or the immune response, suggesting a conserved mechanism of virulence. In mammals the inflammatory response to *C. neoformans* can contribute to host damage [32], while in moths, infection can trigger widespread melanization, which could also damage tissues [33].

The PP and $PP_T$ analysis revealed the importance of comparing results from experiments performed using the same inoculum, especially when comparing the difference in pathogenicity of different strains of the same microbial species, or when comparing a mutant strain to the wild-type. Comparing different PP and $PP_T$ derived from experiments using different inoculum could cause the ΔPP to be off by orders of magnitude depending on the nature of the curve. However, we also demonstrate how pathogenicity data collected using different inocula can be compared by fitting Fs/T versus I plots thus providing new options for comparative analysis. Our results provide support for the view the capacity for virulence is relative, such that labelling a microbe a pathogen under one set of circumstances does not mean the microbe is equally as pathogenic under a separate set of circumstances. PP and $PP_T$ themselves are not intrinsic and immoveable statements on the absolute pathogenicity of a microbe, but rather provide a way to holistically and situationally evaluate pathogenicity given specific factors and variables. The PP would also change in the setting of an infection treated with an effective antimicrobial agent, where the expected Fs and M values decrease, T increases, and I (at the beginning of therapy) remains constant, and as such, changes in PP and $PP_T$ following treatment could be used to measure therapeutic efficacy. Conversely, immunosuppressive treatments or conditions that broadly enhance host susceptibility to infection would lead to increased PP and $PP_T$, which can then be used to identify infection-related the risks involved in certain treatments.

We used published data of *G. mellonella* infection with other microbes to analyze PP vs. I and $PP_T$ vs. I relationships, and found that the linearity of the relationship varied, depending on the microbe. Fungi such as *B. bassiana*, nematode species, *Gm*NPV virus, and some bacteria manifested an exponential negative relationship between PP and I, while some other bacteria, namely *Streptococcus* and *Staphylococcus*, had linear positive relationships between PP and I, indicating that each bacteria contributes directly to pathogenicity in a fixed and measurable amount. Similar trends are seen when we evaluated the $PP_T$ vs. I relationship.

## Development of the Pathogenic Constant $k_{Path}$

The slope of the linear relationship Fs/T and log(I) was defined as $k_{Path}$. The $k_{Path}$ provides a new way describe the relationship between all the components of pathogenic potential

(morbidity, time until onset of mortality, and inoculum) in a manner that is constant at any inoculum and can thus allow for comparisons of pathogenicity between different strains or isolates where the experiments were performed at different inoculum–a comparison that cannot be fairly made using other pathogenic potential metrics. A high $k_{Path}$ would indicate a highly pathogenic microbe, as each additional microbe results in a steep increase in disease and death over time, while a low $k_{Path}$ would indicate a relatively weak microbial pathogen. Additionally, for microbes that cause disease through growth and persistence within tissues, a high $k_{Path}$ could be associated with fast microbial doubling times, whereas low or zero $k_{Path}$ could be associated with microbes that have slower rates of growth within the host. A $k_{Path}$ of zero could also indicate that the microbe is not pathogenic or that the outcome is not dependent on the initial infective inoculum. When this is not the case, as it may not be with *H. capsulatum* or *P. lutzii*, it could indicate that the starting inoculum is irrelevant to disease either because of the presence of a potent toxin that is equally effective in low doses as it is in high doses, or an irregular and slow growth within the host. We note that for some for *H. capsulatum* the values the $k_{Path}$ had a negative sign, which would indicate less severe disease from increasing inocula. While we caution on drawing conclusions from this experimental data until confirmed, it is possible that in some infectious diseases that a threshold inoculum is needed to trigger effective immunity to control infection, which could result in negative $k_{Path}$ values. Additionally, data extracted from mouse literature indicate there is a positive Fs/T vs. I relationship and a positive $k_{Path}$ value for *P. brasiliensis* and *H. capsulatum*. This underscores the importance of comparing data within the same host, and the possibility that the same microbe could have different mechanisms of causing to disease in different hosts. This can then in turn affects the relationship between PP, $PP_T$, and Fs/T versus I relationships. In some microbes, predominantly in bacteria, the relationship between Fs/T and I is linear and not logarithmic. For these microbes, the $k_{Path}$ would be defined differently, and instead rely on the direct inoculum itself. The linear $k_{Path}$ equation could be used to compare bacterial virulence in similar ways between different strains and inoculums. Interestingly, the $k_{Path}$ of *C. neoformans* in *G. mellonella* was nearly the same as it was in mice, again, consistent with the notion that *C. neoformans* behaves similarly in murine and Gallerian host immune systems with regards to virulence. The lines of best fit for PP vs. I, $PP_T$ vs. I, and Fs/T vs. I could be used as a method to roughly predict disease progression and pathogenicity of certain infectious inoculums. This could be helpful for planning experimental design, where a certain disease progression or pathogenicity may be desired for the conditions tested (i.e., antimicrobial drug efficacy during a mild infection).

## Insights into pathogenesis from PP, $PP_T$ and Fs/T versus I relationships

The relationships between parameters of pathogenicity developed here (PP, $PP_T$, Fs/T, $k_{Path}$) provide new potential insights into how the organism cause disease and death within the host. If the microbe has a positive linear relationship in the PP vs I, $PP_T$ vs I, or Fs/T vs I plots, it is consistent with the notion that disease and death primarily result from increasing microbial burden, such that each additional microbial cell causes a proportional increase in host damage that when cumulative would result in the death of the host. This could be a pathogen that damages the host directly through the production of toxic substances or indirectly by eliciting a tissue-damaging inflammatory response that kills the host in a dose-dependent manner or that the host mounts a tissue-damaging inflammatory response that is dependent on microbial burden or a combination of both. The two microbes with the most consistent linear positive relationship were *Staphylococcus aureus* and *Streptococcus* spp., both of which are known to produce a large suite of toxins during infection [34,35]. Conversely, for a microbe that has a

negative exponential PP vs I or $PP_T$ vs I relationship, with a positive logarithmic Fs/T vs I, the magnitude of starting inoculum makes a large contribution to the outcome of the host-microbe interaction and the severity of any ensuing disease. For these microbes, growth and survival in the host determines disease severity, and abundant growth within the host causes death. Microbes that fall under this category included *C. neoformans*, which produces virulence factors such as melanin, polysaccharide capsule, and urease that predominantly allow the fungus to persist and survive within the host rather than intoxicate the host. Consistent with this view, cryptococcosis tends to be a chronic disease that kills the human host after months of slow and progressive damage in the brain, often mediated by increased intracranial pressure resulting from fungal proliferation [36].

In contrast, microbes that produce virulence factors that help survival within the host and damage the host tissues directly (*C. albicans* with candidalysin, adhesins, and proteases), have mixed patterns in their PP, $PP_T$, and Fs/T vs I relationships. *C. albicans* has no clear PP or $PP_T$ vs I relationship, which may be indicative of complex pathogenesis, where it produces a smattering of virulence factors that induce host damage, such as serine aspartyl proteases, candidalysin, and confronts the host with both hyphal and yeast cells [37–42], biofilms, and multiple adhesins [41,43–46]. For *C. albicans*, the mixture of the damage and persistence-type virulence factors could cause no clear PP vs I relationship. *C. albicans* does not have a clear correlation between $PP/PP_T$ and inoculum but does have a positive logarithmic Fs/T vs I relationship, suggests that a mix of host damage and host survival factors may play a role in determining PP and $PP_T$, but the positive logarithmic Fs/T values are determined more by the replication and growth of the fungus within the host.

## Conclusion

Overall, we note remarkable heterogeneity in the relationships between PP, $PP_T$, I, and Fs/T for various microbes with one host, *Galleria mellonella*. We also note that the similarities observed for *C. neoformans* curves with *G. mellonella* and mice suggests commonalities between the interaction of this fungus with a mammalian and insect host, respectively, and hint that certain patterns may be conserved. We consider this study a preliminary exploration of a complex topic, but we note that it is discriminating amongst pathogenic microbes and provides new insights into the problem of virulence. We caution that the results described here involved mostly involved data in the *G. mellonella* host, which lacks an adaptive immune response. Furthermore, we caution that insights gathered from analysis of literature data came from different research groups, which carries the potential for considerable inter-laboratory experimental variation. While we find similarities between PP, $PP_T$, and Fs/T versus I *C. neoformans* infections in murine and Gallerian hosts, a more detailed understanding of the commonalities and differences in host-microbe interactions will require detailed studies in other systems. This is especially the case with human infections, where there is tremendous variability in immune systems, underlying conditions, and environmental variables within the global population that would require nuanced studies and analysis.

In summary, we use the pathogenic potential equations to identify new and unexpected relationships between important variables in the study of microbial pathogenesis such as Fs, I and T. The differences observed here in PP vs. I and Fs/T, imply differences in pathogenesis that are likely to reflect different strategies to survive within the host, promote their own dissemination, and cause host damage over time. For example, if a microbe causes damage through growth and survival, the order of magnitude (log) inoculum would likely be the relevant determining factor of disease (i.e. logarithmic Fs/T vs. I relationship. Whereas if the microbe causes damage through toxins or lytic proteins, pathogenicity would likely be directly dependent upon each microbial cell

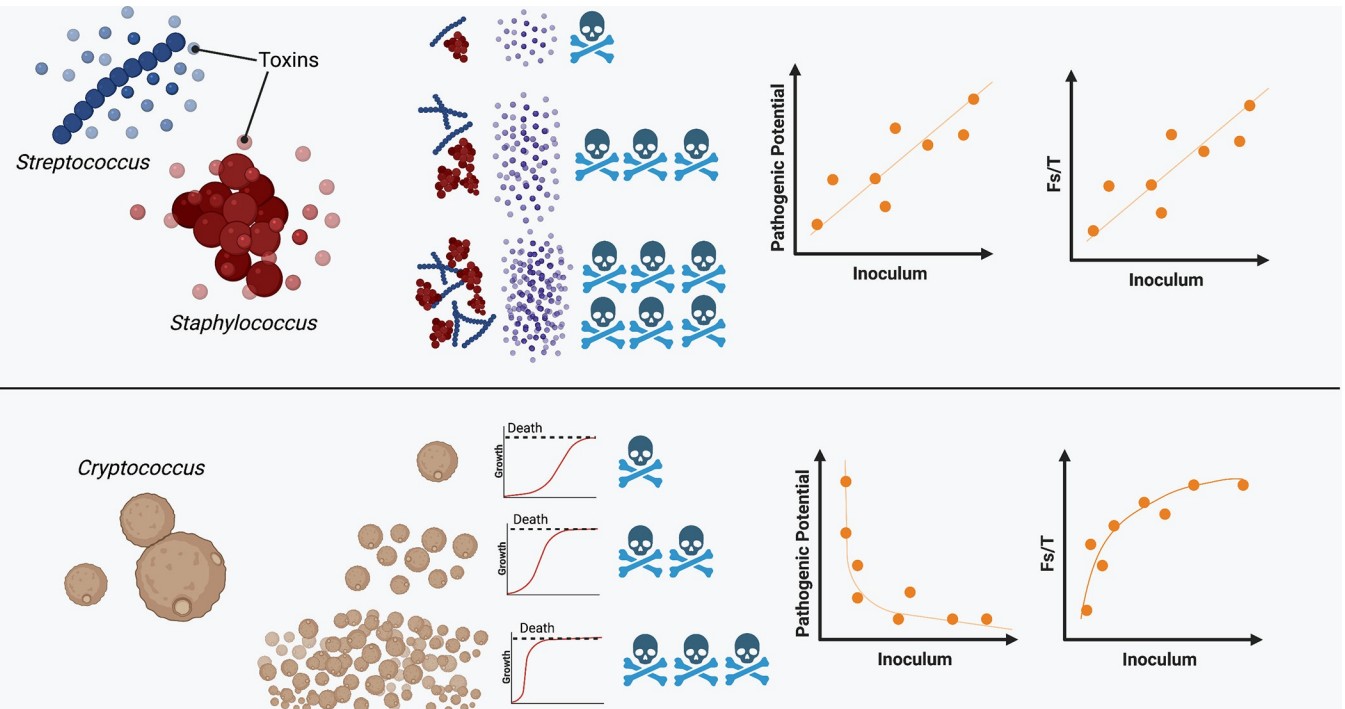

**Fig 10. Model for how differing mechanisms of microbial pathogenesis affect PP vs I and Fs/T relationships.** The top panel indicates microbes, such as *Streptococcus* and *Staphylococcus* that produce toxins that have a dose-dependent effect on survival. This results in positive PP vs. I and a positive linear Fs/T vs I relationships. In the bottom panel is *Cryptococcus* which has an exponential negative relationship with PP vs. I and a logarithmic positive relationship with Fs/T vs. I, which we propose is because *Cryptococcus* causes host death through fungal burden, which would be a log-based relationship between starting inoculum and disease, rather than a dose-dependent linear one.

(i.e. positive linear Fs/T vs. I relationship) (Summarized in Fig 10). Explaining the differences in the shapes and signs of the PP vs. I, $PP_T$ vs. I, and Fs/T curves suggests new avenues for research that could provide fresh insights into the problem of virulence.

## Materials and methods

### Biological materials

*G. mellonella* last-instar larvae were obtained from Vanderhorst Wholesale, St. Marys, Ohio, USA. Cryptococcus *neoformans* strain H99 (serotype A) and *Candida albicans* strain 90028 were kept frozen in 20% glycerol stocks and subcultured into Sabouraud dextrose broth for 48 h at 30˚C prior to each experiment. The yeast cells were washed twice with PBS, counted using a hemocytometer (Corning, New York, USA), and adjusted to the correct cell count.

### Infections of *Galleria mellonella*

Last-instar larvae were sorted by size and medium larvae, approximately 175–225 mg, were selected for infection. Larvae were injected with 10 μl of fungal inoculum or PBS control. Survival of larvae and pupae was measured daily through observing movement with a physical stimulus.

### Literature survey for calculating PP, $PP_T$, and Fs/T for other microbes

We performed a literature search using combinations of the search terms "*Galleria mellonella*," "inoculum," "Kaplan-Meier," "$LT_{50}$," "10^4, 10^5, 10^6," along with the specific name of the

microbe or murine strain we were interested in investigating further. PP, $PP_T$, and Fs/T were calculated from literature that used *G. mellonella* as a model to study various infectious diseases using the following criteria: (1) the survival of at three inoculums were measured for each microbe, (2) the survival data was measured with enough time resolution to see the individual Kaplan-Meier survival curve (3) there was clear data that had overall mortality of the larvae (i.e. an appropriate y-axis to estimate percent mortality), and (4) there was at least a reported $LT_{50}$ (median survival time) or a Kaplan-Meier curve (with the exception of the *Gm*NPV data) in order to calculate the T and Fs values. Overall, we analyzed data from sixteen papers which mostly fit our criteria. There are other examples in literature that could be used, however, many do not test more than three inoculums, have host survival data with insufficient temporal resolution to accurately determine median survival, or do not report median survival time. For the purposes of this paper, Fs value were calculated as the total mortality of larvae or mice since cumulative incidence for the signs or symptoms of infections in *G. mellonella* and mice are often unreported or inconsistent in literature.

## Statistical analysis and regressions

Linear and non-linear regressions were performed using GraphPad Prism Version 8.4.3. Simple linear regressions were used for the linear regressions. Both semi-log non-linear regressions and one-phase exponential decay non-linear regressions were used. Regression method used is described in the figure legend. For some graphs, the 95% confidence interval was plotted, as calculated by the GraphPad Prism software. Equations of the line used for theoretical PP, $PP_T$, and Fs/T values were generated by GraphPad and calculated using Microsoft Excel.

## Supporting information

**S1 Fig. Pathogenic Potential of Paracoccidioides brasiliensis and Histoplasma capsulatum in Murine Hosts.** Using literature values [19–21], we calculated the pathogenic potential (PP), pathogenic potential in respect to time ($PP_T$), Fs/T, lowest pathogenic inoculum (LPI), and $k_{Path}$ for *P. brasiliensis (*A-D*)* and *H. capsulatum* (E-H) in mice. Lines in (D, H) indicate linear regressions. LPI means calculated lowest pathogenic inoculum.
(EPS)

**S1 Table. Spreadsheet with all the Fs, inoculum, T, and M data used for calculating the PP, $PP_T$, and Fs/T for this work.**
(XLSX)

## Acknowledgments

We would like to thank Ella Jacobs and Dr. Quigly Dragotakes for their feedback during the writing process for this manuscript. Fig 10 was made using BioRender.com.

## Author Contributions

**Conceptualization:** Daniel F. Q. Smith, Arturo Casadevall.

**Data curation:** Daniel F. Q. Smith.

**Formal analysis:** Daniel F. Q. Smith, Arturo Casadevall.

**Funding acquisition:** Arturo Casadevall.

**Investigation:** Daniel F. Q. Smith, Arturo Casadevall.

**Methodology:** Daniel F. Q. Smith, Arturo Casadevall.

**Project administration:** Arturo Casadevall.

**Resources:** Arturo Casadevall.

**Supervision:** Arturo Casadevall.

**Visualization:** Daniel F. Q. Smith, Arturo Casadevall.

**Writing – original draft:** Daniel F. Q. Smith, Arturo Casadevall.

**Writing – review & editing:** Daniel F. Q. Smith, Arturo Casadevall.

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
