## [Decision Letter · Decision Letter 0]

3 May 2022

Dear Casadevall,

Thank you very much for submitting your manuscript "On the relationship between Pathogenic Potential and Infective Inoculum" for consideration at PLOS Pathogens. As with all papers reviewed by the journal, your manuscript was reviewed by members of the editorial board and by several independent reviewers. The reviewers appreciated the attention to an important topic. Based on the reviews, we are likely to accept this manuscript for publication, providing that you modify the manuscript according to the review recommendations.

The reviewers positively reviewed this manuscript, but had some comments (including, the data interpretation and clarity of the manuscript) that would require some edits/clarifications to be made in the manuscript.

For readability and clarity, the discussion should focus on the most important points and be more organized.

No additional experiments are necessary.

Sincerely,

Michal A Olszewski, DVM, PhD

Guest Editor

PLOS Pathogens

Sarah Gaffen

Section Editor

PLOS Pathogens

Kasturi Haldar

Editor-in-Chief

PLOS Pathogens

orcid.org/0000-0001-5065-158X

Michael Malim

Editor-in-Chief

PLOS Pathogens

orcid.org/0000-0002-7699-2064

The reviewers positively reviewed this manuscript, but had some comments (including, the data interpretation and clarity of the manuscript) that would require some edits/clarifications to be made in the manuscript.

For readability and clarity, the discussion should focus on the most important points and be more organized.

No additional experiments are necessary.

Reviewer Comments (if any, and for reference):

Reviewer's Responses to Questions

**Part I - Summary**

Reviewer #1: The study by Smith and Casadevall is an interesting approach towards developing a relationship between the pathogenic potential of a microorganism and its infective inoculum. Building on their previous work, the authors develop relationships for the pathogenic potential of fungi and bacteria in Galleria mellonella and compare the results to findings in literature. The result is some interesting quantitative insight in how different microbes cause pathogenicity. Few comments are below.

Reviewer #2: The manuscript by Smith and Casadevall “On the Relationship between Pathogenic Potential and Infective Inoculum” is a refinement of previous papers on a mathematic model of pathogenic potential. In this iteration, the authors extend their studies to incorporate infective inoculum into their calculation.

Overall, this paper adds to the literature of mathematical biology or mathematical microbiology if you will. The work draws upon previously published work as well as new data. The modeling is largely centered on moth larvae but does incorporate some mouse data. There are some concerns about interpretation and conduct of this work.

Reviewer #3: This manuscript is based on the premise that current quantitative measures of pathogenicity are poor, and the goal is to provide an improved quantitative measure. The authors have previously defined a concept called pathogenic potential that is an attempt provides a numerical measure of a pathogen’s virulence that is improved over LD50. The authors define pathogenic potential is a numerical description of an individual microbe, virus, or parasite’s ability to cause disease in a host using the independent variables of inoculum and host, and determining symptomology, mortality and time. Taking the concept of their 2017 paper, they now perform experiments to investigate the relationship between infective inoculum and pathogenic potential of Cryptococcus neoformans in larvae of Galleria moths. The authors establish a new parameter, the pathogen pathogenicity constant or Kpath to quantitatively compare the relative virulence and pathogenicity of a microbe in each host. I have reviewed the mathematic equations and cannot find an error. The authors have appropriately identified the limitations of the study. The authors define different relationships between PP, Fs and I and provide different explanations for the different relationships. The concepts are helpful defining potential host and virulence mechanisms. A lot of thought has been put into the interpretations provided in Table 1 and they are a powerful addition to the concept.

**Part II – Major Issues: Key Experiments Required for Acceptance**

Reviewer #1: (No Response)

Reviewer #2: The authors in Figure 3 propose that one can calculate the smallest inoculum that causes pathogenicity. It would be useful if they would prove that their calculation from the y intercept is accurate.

Reviewer #3: none

**Part III – Minor Issues: Editorial and Data Presentation Modifications**

Reviewer #1: 1. The Discussion is somewhat dense. It would be easier for the reader if the authors could parse out the discussion into several sections. Also, the authors’ logic in referring to Feynman’s terminology in line 305 is unclear. Furthermore, the authors should use the word ‘equation’ instead of ‘formalism’ when referring to equations in lines 60 and 72. Similarly, the authors should not use the word ‘simulate’ in line 247 and Figure 9, as that is commonly used when a computer simulation is performed (which does not appear to be the case here).

2. It appears that the end goal for such a model is to utilize it for hosts with adaptive immunity or in examining drug efficacy. The authors have shown data comparing Fs/T vs. I values for Galleria vs. mouse. Have they tried to examine (or can they comment on) how these plots change when the host is receiving a drug post-infection?

3. The calculation of LPI (line 141), and especially the comparison between intravenous and intranasal is particularly interesting. Have the authors considered the possibility of LPI depending on the stage of growth? For example, microbes in exponential phase are more likely to be lethal than those in the stationary phase. Most studies utilize unsynchronized cultures, making this difficult to estimate, but the answer could be insightful.

4. The mathematical model is quite simple and preliminary. There are no issues with it, it's just very simple. Most likely, this model would need to be modified once the authors go to a host with adaptive immunity. Beside this, it is not the first time the authors have raised the topic of pathogenic potential, thus the novelty of this paper is somehow decreased.

Reviewer #2: 1. It is unclear how the authors calculate an Fs for a moth larva. In the original manuscript in mSphere the definition of fraction symptomatic acknowledged that there is a range. How the authors calculated an Fs for a moth needs to be described.

2. The data with dimorphic fungi is difficult to interpret and hard to believe that the higher inoculum is less pathogenic so to speak. I think if they pore over the data with mice that they will find that is not the case and that the calculations using moths may not be accurate. Moreover, the temperatures used to study dimorphs might cause stress in moths and that is not discussed.

3. One factor that is missing is the doubling time of organisms and how that fits into the equation. We know from human observation that the tempo of disease in humans is often correlated with the doubling time of the organism. For example, pneumococcus causes rapid onset of symptoms whereas Mycobacterium tuberculosis is more insidious. The doubling times of these two bacteria differ substantially. It seems to me to a variable that needs to be acknowledged since the authors do incorporate survival time in their calculations. Perhaps for vertebrates, survival time is not the correct T but time to onset of symptoms.

Reviewer #3: 1. The concept of Kpath (slope of I vs Fs/T) is elegant because of its simplicity. However, by introducing the concept of mortality in addition to symptoms and signs the concept has become more complex and lost some of its elegance. As I will describe below, mortality is a sign of disease and does not need to be a unique variable. The authors either need to do a better job of explaining why it is necessary to incorporate this additional level of complexity or remove it.

2. The authors use the word “symptom”. From JAMA network: “A symptom is a manifestation of disease apparent to the patient himself, while a sign is a manifestation of disease that the physician perceives. The sign is objective evidence of disease; a symptom, subjective.” Since the moths are not reporting their subjective impression of how they feel, moths have “signs” not symptoms. I encourage the authors to define the term Fs as fraction with signs or symptoms.

3. While the authors present an appropriately simple model to test the concept, an important strength is the ability to extend it to all microbes and disease manifestations. Consequently, Fs should be defined by what is most appropriate for the microbe and host. By including mortality, the concept is made more complex and also limiting in its ability to be used in other context. By way of an example, it would be powerful to use the concept to assess pathogenic potential of different strains and species of enteropathogenic bacteria with Fs as manifestations of enterocolitis.

4. Fs is defined in each experiment. In the current experiments the authors used the fraction that has died as the measure of Fs, which is appropriate for these experiments. However, death may not be the most appropriate measure of pathogenicity for different types of microbes. In experiments where another parameter might be used as the sign, that parameter could be used to determine Fs. I would encourage the authors to use Fs and define the symptom or sign (which would include death) or explain why it is necessary to increase the complexity by using a combination of Fs and mortality in pathogens where it is inappropriate to do so.

5. The authors have added a mortality term. The implication is that mortality correlates with a more pathogenic microbe. Death is a complex pathologic process, and while it may be a philosophical point, there are many things that are worse than death. I would encourage the use of the most appropriate Fs for each disease and not to include mortality, or explain why it is necessary to increase the complexity by using a combination of Fs and mortality.

6. Authors provide mathematical descriptions for positive logarithmic and negative exponential relationships. The manuscript would be stronger if an intuitive interpretation were also provided.

7. I would encourage the authors to compare the implications of different relationships. As is, the reader is forced to compare the interpretations in table 1 and to try to determine the differences.

8. Figure 1c is described in the results section, but I was unable to find the figure.

9. Line 316 “i” should be “I”.

PLOS authors have the option to publish the peer review history of their article (what does this mean?). If published, this will include your full peer review and any attached files.

Reviewer #1: No

Reviewer #2: No

Reviewer #3: No

Figure Files:

Data Requirements:

Reproducibility:

References:

---

## [Editor Report · Decision Letter 1]

16 May 2022

Dear Casadevall,

We are pleased to inform you that your manuscript 'On the relationship between Pathogenic Potential and Infective Inoculum' has been provisionally accepted for publication in PLOS Pathogens.

Best regards,

Michal A Olszewski, DVM, PhD

Guest Editor

PLOS Pathogens

Sarah Gaffen

Section Editor

PLOS Pathogens

Kasturi Haldar

Editor-in-Chief

PLOS Pathogens

orcid.org/0000-0001-5065-158X

Michael Malim

Editor-in-Chief

PLOS Pathogens

orcid.org/0000-0002-7699-2064

The authors have addressed all the issues in a satisfactory manner - congratulations on a very interesting contribution.
---

## [Editor Report · Acceptance letter]

1 Jun 2022

Dear Casadevall,

We are delighted to inform you that your manuscript, "On the relationship between Pathogenic Potential and Infective Inoculum," has been formally accepted for publication in PLOS Pathogens.

Best regards,

Kasturi Haldar

Editor-in-Chief

PLOS Pathogens

orcid.org/0000-0001-5065-158X

Michael Malim

Editor-in-Chief

PLOS Pathogens

orcid.org/0000-0002-7699-2064